# Immunology of Multisystem Inflammatory Syndrome after COVID-19 in Children: A Review of the Current Evidence

**DOI:** 10.3390/ijms24065711

**Published:** 2023-03-16

**Authors:** Filippos Filippatos, Elizabeth-Barbara Tatsi, Athanasios Michos

**Affiliations:** Infectious Diseases and Chemotherapy Research Laboratory, First Department of Pediatrics, Medical School, National and Kapodistrian University of Athens, “Aghia Sophia” Children’s Hospital, Athens 11527, Greece; filippat@med.uoa.gr (F.F.); etatsi@med.uoa.gr (E.-B.T.)

**Keywords:** SARS-CoV-2, COVID-19, multisystem inflammatory syndrome in children, humoral immunity, cellular immunity

## Abstract

Immune responses following severe acute respiratory syndrome coronavirus 2 (SARS-CoV-2) in children are still under investigation. Even though coronavirus disease 2019 (COVID-19) is usually mild in the pediatric population, some children exhibit severe clinical manifestations, require hospitalization, or develop the most severe condition: a multisystem inflammatory syndrome in children (MIS-C) associated with SARS-CoV-2 infection. The activated innate, humoral and T-cell-mediated immunological pathways that lead certain pediatric populations to present with MIS-C or remain asymptomatic after SARS-CoV-2 infection are yet to be established. This review focuses on the immunological aspects of MIS-C with respect to innate, humoral, and cellular immunity. In addition, presents the role of the SARS-CoV-2 Spike protein as a superantigen in the pathophysiological mechanisms, discusses the great heterogeneity among the immunological studies in the pediatric population, and highlights possible reasons why some children with a certain genetic background present with MIS-C.

## 1. Introduction

Severe Acute Respiratory Syndrome Coronavirus 2 (SARS-CoV-2), the causative agent of Coronavirus Disease 2019 (COVID-19), rapidly reached pandemic proportions resulting in more than 600 million confirmed cases in all age groups around the world [1]. In the United States of America (USA), there have been more than 14 million COVID-19 cases, according to the Centre for Disease Control and Prevention (CDC) [2]. Although COVID-19 is usually mild in the pediatric population, some children rarely present with severe clinical manifestations. In April 2020, the first reports from the United Kingdom described a clinical presentation in children that shares similar clinical and laboratory characteristics with incomplete Kawasaki Disease (KD) or Toxic Shock Syndrome (TSS) [3]. The condition has been called Multisystem Inflammatory Syndrome in Children (MIS-C), associated with SARS-CoV-2 infection, and there have been many cases documented worldwide [4,5,6]. The same condition is encountered in the literature as Pediatric Multisystem Inflammatory Syndrome (PMIS), Pediatric Inflammatory Multisystem Syndrome temporally associated with SARS-CoV-2 (PIMS-TS), pediatric hyperinflammatory syndrome or pediatric hyperinflammatory shock [7]. 

MIS-C represents a relatively small proportion of the total COVID-19 cases in children. According to the CDC, as of January 2023, there are over 9300 total cases with a median age of 9 years that meet the criteria of MIS-C and 76 reported deaths (estimated mortality of 0.8%) [8]. Compared to the predominance periods of the Alpha, Beta and Delta variants of SARS-CoV-2, it is evident that MIS-C is significantly less frequent and less severe during the Omicron period [9]. The CDC and the World Health Organization (WHO) case definitions of MIS-C have some differences. Both definitions require the presence of fever, elevated inflammatory markers, at least two organ system involvement, evidence of SARS-CoV-2 infection or exposure, and the exclusion of other potential causes [10,11]. The definitions mainly differ in the duration of the fever, as well as hospitalization requirements. The CDC case definition requires that the child must have severe symptoms requiring hospitalization, whereas the WHO case definition does not [10,11].

Even though the timing of symptoms onset in COVID-19 varies per case, the usual MIS-C onset is between 2–6 weeks after acute infection, but rare cases of MIS-C occurring > six weeks after the acute SARS-CoV-2 infection have also been reported [12]. In some cases, the time interval between acute infection and the onset of MIS-C symptoms remains unknown due to asymptomatic acute infection. Interestingly, MIS-C cases seem to increase at least 3–6 weeks after a peak of SARS-CoV-2 transmission within society [13,14]. This time interval coincides with the time of development of acquired immunity and suggests the postinfectious nature of MIS-C. 

The pathophysiology of MIS-C is not fully recognized. It is indicated that MIS-C is the result of an abnormal immune dysregulation of the virus comparable to KD, macrophage activation syndrome (MAS), and cytokine release syndrome. However, it appears to have an immunophenotype that is distinct from KD and MAS [15,16]. 

In this review, the innate and adaptive immunological aspects of MIS-C are discussed based on current evidence, and the great heterogeneity between SARS-CoV-2 immunological studies and the genetic background of patients with MIS-C are discussed. 

## 2. Incidence of MIS-C

Nowadays, MIS-C has a worldwide distribution, but there are notable differences among different world regions and different SARS-CoV-2 variants. As by the end of February 2023 in the USA, the highest incidence of MIS-C in 50 different jurisdictions was encountered in California, followed by Texas, Florida, Georgia, Ohio and New York, while in other states, such as Oklahoma or Nevada, reported cases were relatively lower [8].

Data from the USA, Europe, Asia and Africa indicate a lower incidence of MIS-C during Delta and Omicron variant predominance periods. In the USA, MIS-C incidence and complications were significantly reduced during the Delta period compared to previous pandemic waves [17]. The same trend was also confirmed in Greece, with the estimated incidence of MIS-C being notably reduced from 3.5/1000 in Alpha to 0.25/1000 in the Delta predominance period [18]. After the onset of Omicron, the incidence and severity of MIS-C was even lower compared to Alpha and Delta variant waves in Israel [19], while in a study in South Africa in November 2021, no patients who fulfilled WHO criteria for MIS-C diagnosis were encountered [9]. It is possible that MIS-C epidemiology has remarkably changed as a consequence of widespread exposure of the pediatric population to successive COVID-19 waves, particularly the Omicron variant, mainly characterized by increased transmissibility or reinfection rates. 

Pre-existing SARS-CoV-2 immunity, either as a result of natural infection or vaccination in children, seems to play a crucial role in MIS-C pathogenesis and severity. According to CDC, SARS-CoV-2 seroprevalence attributed only to infection-induced immunity has reached 92% in children by March 2023, while a significant increase in seropositivity during Omicron compared to Delta variant in the pediatric population has also been observed in Europe as well [20]. However, SARS-CoV-2 vaccination has also affected MIS-C incidence. During the Omicron wave, a study by Holm et al. showed that the incidence of MIS-C was considerably reduced among vaccinated children in contrast with individuals who had not been vaccinated (one vaccinated vs. 11 unvaccinated MIS-C patients) [21].

## 3. Innate Immune Responses in MIS-C

Overexpression of certain innate immunological pathways might lead not only to MIS-C onset but also to increased MIS-C severity and complications. These pathways do involve not only innate immune system cells and inflammatory markers but also cytokines, chemokines and certain gene activation. The most important immunological factors MIS-C regarding innate immune responses in MIS-C are presented in Table 1. 

MIS-C is a severe condition with abnormal blood cell counts, elevated inflammatory markers, and pro-inflammatory mediators disturbances, some of the most common characteristics during the onset of the syndrome [15]. Lymphopenia, neutrophilia, and elevated inflammatory markers such as C-reactive protein (CRP), erythrocyte sedimentation rate (ESR), D-dimers, procalcitonin, fibrinogen and ferritin are detected in most patients with MIS-C and correlate with disease severity [22,23]. Among patients with MIS-C, a higher mean CRP value (32.1 mg/dL, normal values: 0–0.5 mg/dL) was observed in those who developed shock, as was an increased neutrophil count (16 × 10^9^/L, normal values: 1.5–7 × 10^9^/L) and a decrease in lymphocyte count (0.7 to 1.3 × 10^9^/L, normal values: 1.5–4 × 10^9^/L) in those who did not [22]. Thrombocytopenia and low eosinophil counts are the main distinguishing characteristics of MIS-C compared to KD and appear to be related to the degree of viral pandemic (ViP) signatures activation, the levels of IL-15 and MIP1 expression, which is a critical component of the immune system component [24]. Compared to Kawasaki, patients with MIS-C have significantly lower absolute neutrophil, lymphocyte, platelet, and eosinophil values [24]. Innate immunological differences encountered in MIS-C compared to Kawasaki disease are presented in Table 2.

Neutrophils, the most abundant circulating phagocytes, are stimulated by several major chemoattractant factors, such as complement-derived C5a and the neutrophil chemokine interleukin-8 (IL-8) [55]. In pediatric patients with COVID-19, there is interferon-mediated gene stimulation in neutrophils [26]. In a study by Seery et al., a negative association of neutrophils with CD11b, CD66b, and L-selectin was observed, while CD64 in neutrophils was a marker of disease severity as it varied significantly between asymptomatic and symptomatic COVID-19 children [25]. The expression of CD11b, CD66b, LAIR-1, and PD-L1 in neutrophils is significantly higher in MIS-C patients [25]. In MIS-C, there is a notable up-regulation of the *CCRL2 ELMO2*, *GPR84*, *IRF7*, *IFIT3*, and *MX1* genes, which are responsible for neutrophil chemotaxis and are enrolled in multiple immunological pathways, including type I and II interferon activation or β2-integrin stimulation [26]. However, the expression of antigen-presenting cells is relatively reduced [27]. It is possible that MIS-C patients may be characterized by neutrophil extracellular traps (NET), a condition commonly encountered in adults with moderate and severe COVID-19 associated with thrombosis and endothelial damage [28,29]. Seery et al. showed that there are no significant differences between children with COVID-19 and MIS-C in NETs [25]. Further studies in the field of NETs are required in order for a pathophysiological connection with MIS-C to be revealed. 

The role of NK cells in the pathogenesis of MIS-C remains elusive. Current evidence supports the increased expression of genes, such as *CCL4*, which are associated with increased cytotoxic potential and impairment in peripheral tissues [27]. There is a positive correlation between IFN-gamma levels in plasma with NCR1, the soluble marker of NK cells [31]. 

In a study by de Cevins et al., NF-kB signaling, VEGF signaling, and inflammatory pathways (type I and type II IFNs, IL-1, IL-6, and IL-17 signaling pathways) were substantially outnumbered in patients with MIS-C [30]. A sampling of monocytes and dendritic cells from MIS-C patients revealed a significant decrease in the expression of NF-kB inhibitors, including NFKBIA, NFKBID, NFKBIE, and NFKBIZ. In MIS-C without myocarditis, the expression of TRAIL, IL-7, IL-2, IL-13, IFN-g, IFN-a2, IL-17A, and Granzyme B was slightly enhanced compared to MIS-C with myocardial involvement [30]. A modification of type I and type II IFN signaling pathways, alongside overexpression of many IFN-stimulated genes (ISGs) (*JAK2, STAT1, STAT2, IFITM1, IFITM2, IFI35, IFIT1, IFIT3, MX1, IRF1*) in dendritic cells and monocytes was also shown in patients with MIS-C without myocarditis exclusively [30]. Increasing VEGF, TGF-a, and TGF-b may constitute potential mediators of angiogenesis and vascular homeostasis, while elevated chemokines (CCL2, CCL3, CCL20, CX3CL1, and CX10; CCL2, CCL3, CCL20, CX3CL1, and CX10, respectively) may facilitate enhanced cell migration toward inflamed sites [30]. Inflammatory cytokines, such as TNF-a, TGF-b, IL1b, IL-13, IL-4, and VEGF, are possible to enhance cardiac fibroblasts’ development into cardiac myofibroblasts [30]. 

Cytokines and chemokines play a vital role in the initiation, prolongation or downregulation of the immune response in the COVID-19 in the pediatric population. In MIS-C, certain cytokines, such as TNF, IL-1β, IFN-gamma, IL-6, IL-8, IL-10 and IL-17 and chemokines, such as CCL2, CCL3, CCL4, CXCL1, CXCL5, CXCL6, CXCL9, CXCL10 or CXCL11, can be elevated compared to pediatric patients with uncomplicated COVID-19 [15,16,27,32,33,34,35]. Plasma IL-6 was elevated in patients with severe MIS-C, but levels were not superior when compared to IL-6 levels in pediatric septic patients [56]. Apart from IL-1 and IL-8, which were not significantly elevated in MIS-C compared to KD, there is a robust enhancement of TNF-a, IFN-gamma and IL-10 production in MIS-C compared to KD (Table 2) [24]. The inflammatory mediator IL-17 plays a more prominent role in the progress of KD than in MIS-C (Table 2) [53], while cytokine profiles demonstrated that patients with severe MIS-C had higher levels of circulating IFN compared to those with KD (Table 2) [54]. It is important to state that the levels of cytokines and chemokines may significantly vary among studies that use different proportions of age groups, the timing of sampling or diagnostic methods. Patients with potentially fatal COVID-19 were found to carry uncommon mutations in at least 13 loci that result in compromised IFN pathways, according to research by Zhang et al. [57].

Patients with MIS-C experience a higher rate of thrombotic events than those with mild to moderate disease, and a possible role of the complement system has been implied [58]. The complement system may play a key role in the development of MIS-C since soluble C5b-9 in the plasma of pediatric patients with MIS-C and uncomplicated COVID-19 are significantly higher than those of healthy controls [36]. Due to its carbohydrate-residue-rich surface structures, SARS-CoV-2 may play a significant role in the lectin complement pathway in the pathogenesis of diseases related to the virus [59]. Preliminary data from plasma samples proteomics analysis suggest that phospholipase A2 (PLA2G2A) may help as a biomarker for the diagnosis of thrombotic microangiopathy in MIS-C [31]. Abnormally high levels of soluble C5b-9, possibly as a result of the augmented autoreactivity observed in MIS-C, indicate complement activation and endothelial damage, thus making it a useful biomarker for tracking complement terminal sequence activity [36]. 

Early recognition of MIS-C is crucial, and there are several inflammatory biomarkers that could distinguish MIS-C from Kawasaki disease. For instance, higher IFN-gamma CXCL9 values cannot only distinguish MIS-C from Kawasaki patients but are also associated with multiple organ dysfunction, including kidneys or cardiovascular system [60]. 

## 4. Humoral Immune Responses in MIS-C

Inflammatory responses against a specific pathogen are driven by interactions between several viral and host parameters, but in the case of MIS-C, the evidence mentioned above suggests that viral factors are less likely to explain why some children develop MIS-C while others do not [61]. Humoral immune responses in MIS-C involve total antibodies, IgG, IgM, and IgA, against SARS-CoV-2 Spike protein or other SARS-CoV-2 proteins or autoantibodies and may also play a key role in a certain subtype of MIS-C in neonates: Multisystem Inflammatory Syndrome in Neonates (MIS-N). The most important immunological factors MIS-C regarding humoral immune responses in MIS-C are presented in Table 1. 

Although the majority of patients have detectable SARS-CoV-2 antibodies in MIS-C onset, approximately 30% of them have positive reverse transcription PCR (RT-PCR) test on a nasopharyngeal swab [62,63] and, compared to severe COVID-19, their cycle thresholds are higher [32]. The decreased SARS-CoV-2 viral load detected in the upper respiratory tract of MIS-C patients is indicative of the highly effective innate immune responses of children during the acute phase of the disease that enables the early elimination of the virus in the upper airways [61]. 

Possibly, host factors also play a key role in the immune dysregulation in MIS-C, and future research is required. In contrast, there are several studies describing the prolongation of SARS-CoV-2 detection in organs often involved in MIS-C, such as the gastrointestinal tract, kidneys, and cardiovascular or central nervous system [64]. 

Quantitative SARS-CoV-2 serology assists in distinguishing MIS-C from acute COVID-19 since higher titers are observed in MIS-C [37]. The onset of antibody responses in children with COVID-19 is found to be negatively associated with disease severity [65]. Due to the delayed onset of MIS-C, the levels of IgM are found to be lower compared to IgG [38]. 

In MIS-C, IgA antibody titers do not show significant differences between acute and convalescent phases of the disease and are associated with gastrointestinal clinical manifestations [33]. However, IgA levels have been shown to be higher in MIS-C compared to pediatric patients [39]. The role of elevated circulating IgA antibody levels in the prevention of severe COVID-19 has well been established in hospitalized adult patients [66,67]. Gruber et al. studied mucosal immunity in SARS-CoV-2 in 9 MIS-C patients [33]. IgA antibody titers remained increased in MIS-C patients during convalescence, like the acute phase, while L-17A activation and mucosal chemotaxis via CCL20 and CCL28 stimulation were also noted [33]. 

Humoral immune responses largely depend on the structural proteins of SARS-CoV-2, and the magnitude of antibodies elicited after natural infection is interpreted accordingly. Two of the most important proteins are encoded in the SARS-CoV-2 genome: the S protein, which is the main antigenic target of cytotoxic lymphocytes that also stimulate neutralizing antibodies, and the nucleocapsid (N) protein which is responsible for viral RNA replication [68]. When comparing antibody responses against the S protein (anti-S) and the N protein (anti-N) of SARS-CoV-2 in MIS-C patients, current evidence demonstrates the predominance of anti-S IgG and a limited antibody repertoire, with enhanced ability to activate monocytes [33,39,40]. This contradicts adult SARS-CoV-2 patients who express a broader antibody repertoire of anti-S IgG, IgM, IgA and anti-N IgG with additionally increased neutralizing activity [39]. Anti-S IgG, anti-S IgM and anti-N IgG levels may not significantly vary between children with and without MIS-C but are both significantly lower when compared to adults with severe COVID-19 [39]. Further investigation into the mechanisms that underline the low-prevalence anti-N IgG with decreased viral replication in the onset of MIS-C is required. 

Bartsch et al. showed that children with severe MIS-C can develop high levels of inflammatory monocyte-activating SARS-CoV-2 IgG antibodies that are retained, and this can distinguish them from children without MIS-C [40]. According to current evidence, neutrophils and monocytes of MIS-C patients may have an increased expression of CD64 which has enhanced affinity with the Fc fragment of IgG [33]. The selective extension of certain IgG subtypes in MIS-C has not been well understood. 

Except for antibodies produced after natural infection, autoantibodies are also frequently detected in MIS-C patients, but their role in disease progression remains unknown. Bastard et al. discovered a disrupted Type I IFN signaling pathway in 10% of adult patients with life-threatening SARS-CoV-2 infection due to autoantibodies against Type I IFNs, particularly IFN-a, which resulted in the immune system’s inability to combat the viral infection [41]. Autoantibodies against IL-1Ra or other autoimmune conditions, such as systemic lupus erythematosus, Sjögren’s syndrome or autoimmune gastritis, were detected in more than half of MIS-C patients recruited in studies by Pfeifer et al. and Burbelo et al. [42,43]. It is not clear whether the administration of intravenous immunoglobulin (IVIG) or blood product transfusions may interfere with the detection of autoantibodies in serum and cause misleading results. There are certain IgG and IgA autoantibodies possibly attributed to exposure to antigens from the gastrointestinal tract or mucosa [27,33], which could partially explain the autoreactivity and immune dysregulation encountered in MIS-C. Since most of the patients have a good response to immunomodulatory therapy with IVIG [69], a possible underlying mechanism regarding the activation of certain IgG immune complexes in MIS-C pathogenesis could be implied [14].

MIS-N from maternal SARS-CoV-2 exposure is a rare condition that is often associated with cardiovascular complications, respiratory distress, or gastrointestinal manifestations [70]. Recent case series describe that the usual onset of MIS-C is approximately one month after the onset of maternal COVID-19 onset [71]. This coincides with the time-point of the peak of IgG maternal antibody levels in serum and the umbilical cord blood [72,73]. However, the highest transfer is achieved when the onset of infection is 2–6 months before labor [73]. Consequently, it remains unclear whether the magnitude of maternal IgG antibody levels is a predominant factor for MIS-N development. Current evidence on the connection between the transplacental transfer of maternal natural infection antibodies and the onset of the condition requires further investigation.

## 5. Cellular Immune Response in MIS-C

The most important immunological factors MIS-C regarding cellular immune responses in MIS-C are presented in Table 1. Lymphopenia caused by SARS-CoV-2 results from a combination of the virus’s ability to directly induce apoptosis of lymphocytes, as well as its suppression of the thymus, the impairment of bone marrow function, and its ability to cause a redistribution of T lymphocytes in peripheral blood circulation, resulting in damage to affected organs [74]. Increased inflammatory markers, intense severity of radiographic abnormalities in the lungs, and a poor clinical outcome were all linked to the redistribution of dendritic cells, monocytes, and lymphocytes (naïve, memory or effector cells) from peripheral tissues to the lower respiratory system [75,76]. Lymphopenia is a marker of severe disease in children with COVID-19 and is relatively common in MIS-C. By causing lymphopenia and exhaustion in T cells, including gamma-delta T-cells and CD8^+^ T-cells, SARS-CoV-2 can suppress its activity, therefore leading to a major malfunction in the immunoregulatory mechanism of the adaptive immune system’s immunoregulatory mechanism [44,45,46]. Although the study by Ramaswamy et al. showed increased proliferation of CD4+ T cells in MIS-C patients compared to healthy controls, other studies support that lymphopenia in patients with MIS-C affects both CD4^+^ and CD8^+^ and not certain functional categories [27,33,45].

MIS-C induces the proliferation of plasmablasts, as well as the activation of B cell clones, while elevated plasmablast counts are also observed in children with uncomplicated COVID-19 [45]. Increased rates of plasmablasts expressing the T box transcription factor T-bet also suggest extrafollicular responses in patients with MIS-C [45]. To better understand what guides this activation of B cells in moderate and severe COVID-19 in children, research should focus more on comparing the characteristics of expanded B cells and plasmablasts in the future.

Characterizing the TCR repertoire of MIS-C patients, Porritt et al. discovered a significant overexpression of the TRBV11-2 gene, which was linked with disease severity cytokine production and detected a remarkable concordance in the distribution of HLA-class I alleles A02, B35, and C04 among those patients [47]. In MIS-C, there is a predominance of CCR6 in CD4+ T cells, while cross-reactive CD4^+^ and CD8^+^ SARS-CoV-2 T cells have also been detected with other common cold coronaviruses [77].

The functional phenotype of an activated T cell is highly susceptible to the conditions under which the cell was stimulated [78]. Differentiation to these functional T cell subsets is affected by several factors, including the amount of antigen, the route by which antigens enter the body (e.g. upper respiratory or gastrointestinal tract), and the type of antigen (viral or bacterial) [78]. In contrast to the predominant Th1, CD25^+^, and IFN inflammatory response in adults, children are characterized by a particularly noteworthy Th2 and Th17 immune response [79]. Th1 cell development is primarily regulated by IL-12 and IFN-gamma, while Th2 cell development is largely determined by IL-4 [78].

CD8^+^ T cells in MIS-C show increased signatures of cytotoxicity compared to those in healthy children, though the differences are not as remarkable as they are in NK cells [47]. Abnormal depletion of NK cells and the exhaustion of effector CD8^+^ T cells can contribute to chronic inflammation, as revealed by RNA sequencing in children with MIS-C [80]. Compared to pediatric COVID-19 patients, no significant differences were observed in the subsets of naive CD4+ and CD8^+^ naïve, central memory and effector memory lymphocyte subsets [45]. The activation of CX3CR1^+^CD8^+^ T cells is another distinguishing characteristic of MIS-C compared with pediatric COVID-19, which may possibly have consequences in the development of cardiovascular complications [45]. In patients with MIS-C, the percentage of activated CX3CR1^+^CD8^+^ T cells decreases as the clinical condition improves, a finding that has important implications for the disease [45]. Therefore, it appears that CD8^+^ T cells, especially those that express CX3CR1, are persistently activated and dysregulated [45]. Activated CX3CR1+ CD8+ T cells and activated HLA-DR+ CD38+ or Ki67+ CD8^+^ and CD4^+^ T cells are found to be more abundant in MIS-C than in patients with severe pediatric COVID-19, but HLA-DR+CD38+ CD8^+^ T cells frequencies decreased in the first two weeks of admission in MIS-C [45]. A study in MIS-C patients which was based on peripheral blood mononuclear cells (PBMCs) isolation after fresh, non-defrost whole peripheral blood collection from patients before pharmaceutic immunosuppression showed that CD8^+^ T cell-mediated IFN-gamma release was found to be significantly higher in children with MIS-C than in hospitalized children one month after acute SARS-CoV-2 infection [48]. 

T-cell responses play an important role in MIS-C and involve CD4^+^ and CD8^+^ lymphocyte activation, as well as lymphocyte-produced cytokines. However, due to the different immunological assays that are used to investigate cellular immune responses, the results of currently published studies should be interpreted cautiously. 

## 6. SARS-CoV-2 Spike Protein as a Superantigen in MIS-C

Superantigens engage directly with the invariant region of the class II MHC molecule without requiring any process by antigen-presenting cells [81]. An antigen is taken up by an antigen-presenting cell, processed, expressed on the cell surface in association with class II MHC, and recognized by an antigen-specific T-cell receptor during normal T-cell recognition (Figure 1) [81]. 

This mechanism has been well described in TSS attributed to *S. aureus*, in which staphylococcal enterotoxins behave as superantigens capable of activating a large number of T cells and resulting in massive cytokine production [81]. There are two domains of TSST-1 toxin that behave as superantigens and are of major importance for MHC class II binding [82].

The SARS-CoV-2 RNA genome encodes 29 structural and nonstructural proteins, the most important of which are the structural spike (S), envelope (E), membrane (M), nucleocapsid (N) and the non-structural polyprotein ORF1a/b [68]. The S protein is highly glycosylated and contains a single transmembrane domain oriented in the extracellular space [83]. The S protein is divided into two functional parts S1 and S2, where the S1 subunit catalyzes viral adhesion to the cell membrane and the S2 subunit catalyzes fusion [84]. From the amino-terminal to the carboxy-terminal end of the S protein, there are several different functional domains which are in the following order: The amino-terminal domain (N-terminal domain—NTD), the receptor binding motif (RBM) containing the receptor binding domain (RBD), the furin cleavage site (furin cleavage site), the fusion peptide (FP), the central helix (CH), the connecting domain (CD), the heptad repeat HR1/2 domain, the transmembrane domain (TM) and the cytoplasmic tail (CT) [85]. The main antigens that stimulate neutralizing antibodies, as well as important targets of cytotoxic lymphocytes, are located in the S protein and specifically in the RBD domain [68]. 

SARS-CoV-2 viral spike (S) protein may act as a superantigen, causing a cytokine storm that leads to the development of MIS-C, a condition that shares comparable characteristics with TSS (Figure 1) [86]. 

The S protein has a motif with a high affinity for binding TCR [49]. This motif is structurally comparable to staphylococcal enterotoxin B, which is a superantigen that promotes TSS by engaging with TCR and class II molecules [49]. According to simulations, changes in the binding area of the SARS-CoV-2 S protein could alter its ability to engage with MHC class II molecules and TCR [49]. Modeling and analysis have shown that the SARS-CoV-2 virus encodes a superantigen motif close to the S1/S2 cleavage site that interacts with the TCR, as well as the CD28 [49]. In MIS-C patients, HLA class I alleles A02, B35, and C04 were found to be related to TRBV11-2 (V21.3), a skewing of the TCR repertoire in a group of CDR3-independent, indicating a superantigen-mediated activation of T cells [47]. Enhancement of TRBV11-2 in T cells has been shown to be supported by other studies [50,51] despite the fact that differential expression of specific genes involved in superantigen theory has not been elucidated. The superantigen-like motif of the SARS-CoV-2 S glycoprotein was found to correlate significantly with polyacidic residues in the V chain encoded by TRBV11-2 (V21.3), suggesting that the SARS-CoV-2 protein may directly mediate the extension of TRBV11-2 [47]. Novel Rivas et al. also support the hypothesis that SARS-CoV-2 as a superantigen may be responsible not only for MIS-C onset and immune dysregulation but also for prolonged clinical manifestations, such as post-COVID syndrome and complications from the central nervous system [52]. 

In contrast to the rapid disease onset and evolution of cytokine storm seen in TSS, MIS-C is typically observed some weeks after primary SARS-CoV-2 infection [87]. During the acute phase of inflammation, SARS-CoV-2 is usually undetectable in patients with MIS-C; hence, the connection between superantigen theory and MIS-C has yet to be proven. SARS-CoV-2 is constantly evolving as an RNA virus, and more research is needed to determine whether any of the future variants for concern of the virus triggers high inflammatory signaling in immunological and endothelial cells, contributing to MIS-C.

## 7. MIS-C and COVID-19 Vaccination

There is a possibility that the incidence of MIS-C was decreased not only because of substantial exposure of the pediatric population to multiple waves of COVID-19, particularly the Omicron variant, but also due to the significant increase in the percentage of children of all ages who are currently vaccinated for COVID-19 [21]. Risk of exposure, reinfection and severe disease with the infection must be weighed against the unclear safety of vaccination in deciding whether to vaccinate someone who has had SARS-CoV-2 infection followed by multisystem inflammatory syndrome (MIS). Children are also at risk of developing rare but notable immune-related adverse effects after vaccination; this is especially concerning given the theory that MIS-C is related to immunological dysregulation triggered by SARS-CoV-2 infection. Rare case reports of MIS in adults after vaccination underscore the need to monitor this potential side effect [88], although vaccine trials in this age group have not detected any relevant indication. 

There is some indication that vaccination might provide protection against MIS-C. In France, 7/102 patients who received the first COVID-19 vaccine dose developed MIS-C approximately one month after vaccination, with the hazard ratio of MIS-C presentation being estimated at 0.09 [89]. In another study, the majority (95%) of 102 hospitalized adolescents and young adults with MIS-C were not vaccinated [90]. However, none of the five patients with MIS-C who were vaccinated with BNT162b2 required respiratory or cardiovascular assistance in the intensive care unit, with BNT162b2 estimated effectiveness against MIS-C reaching 91% [90]. However, MIS-C after vaccination can be associated with elevated inflammatory biomarkers, such as CRP and D-dimers, as well as cardiac involvement, including elevated Troponin or NT-BNP values or pericardial effusion [91].

According to CDC, if a person has recovered clinically from MIS-C, is at increased risk of exposure to SARS-CoV-2 and MIS-C was not associated with COVID-19 immunization, then the advantages of vaccination may exceed the risks and, therefore, vaccination is suggested [92].

## 8. Genetic Factors Associated with MIS-C

Despite the unprecedented increase in pediatric cases of COVID-19, the overall prevalence of MIS-C remains low [1,8]. Even though humoral immune responses between pediatric COVID-19 and MIS-C do not show significant differences, the onset of MIS-C only to a small subgroup of COVID-19 children could possibly imply an underlying genetic predisposition, including polymorphisms or certain mutations [93]. There appears to be a disproportionate impact of MIS-C in black and Hispanic children in contrast to KD, which has a considerably higher incidence in Asian children [94,95]. However, this finding can at least partially be attributed to the increased risk of SARS-CoV-2 infection in geographic areas with impaired socioeconomic status, without excluding the connection to genetic differences [96]. 

The susceptibility and severity of SARS-CoV-2 disease can be determined by genetic variation among the three major class I histocompatibility complex (MHC) genes: Human Leukocyte Antigen A (*HLA-A*), *HLA-B*, and *HLA-C* genes. The *HLA-B*46:01* genotype suggests an increased susceptibility to SARS-CoV-2 infection. The highest potential for protective cellular immunity coincides with *HLA-B*15:03* genotype presence [97]. 

The importance of the IFN pathway to the clinical manifestations and possible treatments of COVID-19 has been the subject of numerous studies [61]. Patients with potentially fatal COVID-19 were discovered to exhibit rare variations in 13 loci that contribute to loss of function and, thus, defective IFN pathway, according to a study conducted by Zhang et al. [57]. In addition, Bastard et al. discovered that 10% of adult patients with potentially fatal SARS-CoV-2 infection had a disrupted Type I IFN signaling pathway attributed to the presence of autoantibodies against certain Type I IFNs subtypes, such as IFN-a [41]. The Suppressor of Cytokine Signaling 1 (SOCS 1) is an important negative regulator of type I and type II IFN signaling [98]. Patients with SOCS1 haploinsufficiency exhibit elevated STAT1 phosphorylation and a transcriptional signature suggestive of elevated IFN signaling and apoptosis [99]. T-cell activation and significant levels of IFN signaling were observed in a study of two patients with MIS-C who presented with autoimmune thrombocytopenia and hemolytic anemia [99]. Individuals with silencing mutations may be predisposed to infection-related hyperinflammatory conditions such as MIS-C.

## 9. Conclusions

Children rarely experience severe symptoms, require hospitalization, or develop complications such as MIS-C. Since the beginning of the pandemic, children with complicated COVID-19 or MIS-C have been a challenging immune response study group with numerous unexplained immunological aspects. Unspecified innate, humoral, and T-cell-mediated immune mechanisms that predispose certain pediatric subgroups to develop MIS-C or remain totally asymptomatic after SARS-CoV-2 infection are still under investigation. Further research is required, and the great existent heterogeneity of immunological studies should be diminished to better address the clear pathophysiological mechanisms of MIS-C. 

## Figures and Tables

**Figure 1 ijms-24-05711-f001:**
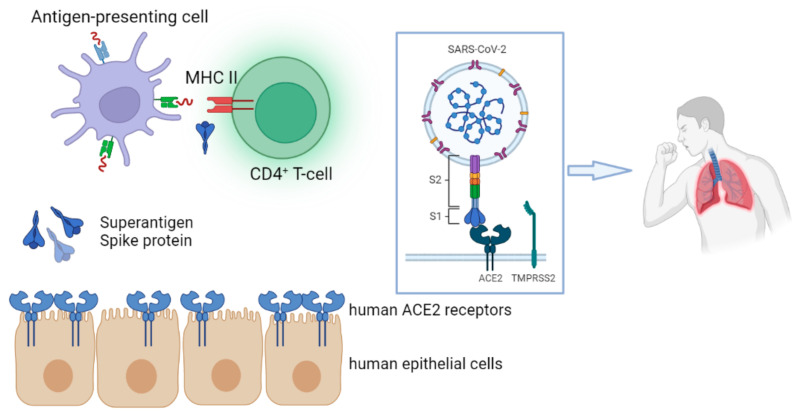
The spike (S) protein of SARS-CoV-2 may act as a superantigen, triggering a cytokine storm that contributes to the development of MIS-C. Superantigens interact directly with the invariant region of the class II MHC molecule, bypassing the need for an antigen-presenting cell-mediated phase. The figure was originally designed by the Biorender in silico tool (https://biorender.com, accessed on 14 January 2023).

**Table 1 ijms-24-05711-t001:** Immunological factors associated with MIS-C in innate, humoral and cellular immunity, as well as superantigen hypothesis regarding the syndrome onset.

Immune System Aspects	Immunological Factors in MIS-C	References
Innate immunity	Lymphopenia, neutrophilia, elevated inflammatory markers, thrombocytopenia and low eosinophil counts	[22,23,24]
	Increased expression of *CD11b*, *CD66b*, *LAIR-1*, and *PD-L1* in neutrophils	[25]
	*CCRL2*, *ELMO2*, *GPR84*, *IRF7*, *IFIT3*, and *MX1* genes upregulation	[26]
	Reduced antigen-presenting cells	[27]
	Increased neutrophil extracellular traps	[28,29]
	Increased signaling pathways (NF-kB, VEGF, IFN, IL-1, IL-6, and IL-17)	[30]
	NK cells: increased expression of *CCL4*, *NCR1*	[27,31]
	Evelated cytokines (TNF, TRAIL, IL-7, IL-2, IL-13, IFN-g, IFN-a2, IL-17A, Granzyme B, IL-8, IL-10)	[15,16,27,30,32,33,34,35]
	Elevated chemokines (CCL2, CCL3, CCL4, CXCL1, CXCL5, CXCL6, CXCL9, CXCL10, CXCL11, CX3CL1, CX10, CCL20)	[15,16,27,30,32,33,34,35]
	Increased expression of IFN-stimulated genes in dendritic cells and monocytes: *JAK2*, *STAT1*, *STAT2*, *IFITM1*, *IFITM2*, *IFI35*, *IFIT1*, *IFIT3*, *MX1*, *IRF1*	[30]
	Increased phospholipase A2 (*PLA2G2A*)	[31]
	Increased soluble C5b-9	[36]
Humoral Immunity	Higher SARS-CoV-2 and IgA antibody levels, lower IgM compared to non-MIS-C patients	[37,38,39]
	Higher SARS-CoV-2 anti-S IgG compared to anti-N IgG levels in MIS-C	[33,39,40]
	Comparable SARS-CoV-2 anti-S IgG, anti-S IgM and anti-N IgG levels to non-MIS-C patients	[39]
	Increased autoantibodies against Type I IFNs, IL-1Ra	[41,42,43]
Cellular immunity	Increased CD4+ and CD8+ T cells	[44,45,46]
	Increased B-cell plasmablasts	[45]
	Increased CD8+ T cells cytotoxicity	[47]
	Increased CX3CR1+CD8+ T cells	[45]
	Increased CD8+ T cell-mediated IFN-gamma	[48]
Superantigen	High affinity of SARS-CoV-2 Spike protein to TCR	[49]
	Εnhancement of TRBV11-2 in T cells	[47,50,51,52]

Legend: CCL: chemokine C-C Motif Chemokine Ligand; CXCL: chemokine (C-X-C motif) ligand; IFN: interferon; IL: interleukin; MIS-C: a multisystem inflammatory syndrome in children; NF-Kb: Nuclear factor-kB; SARS-CoV-2: severe acute respiratory syndrome coronavirus 2; TCR: T-cell receptor; TRAIL: tumor necrosis factor-related apoptosis-inducing ligand; VEGF: vascular endothelial growth factor.

**Table 2 ijms-24-05711-t002:** Innate immunological differences encountered in MIS-C compared to Kawasaki disease.

Immunological Factors	MIS-C	Kawasaki Disease	References
Absolute neutrophil counts	↓	↑	[24]
Absolute lymphocyte counts	↓	↑	[24]
Absolute platelet counts	↓	↑	[24]
Absolute eosinophil counts	↓	↑	[24]
IFN-gamma CXCL9 values	↑	↓	[53]
IL-1 and IL-8	↑	↑	[24]
TNF-a, IFN-gamma and IL-10	↑	↑	[24,54]
IL-17	↑	↑	[53]

Legend: ↓: reduction, ↑: increase; CXCL: chemokine (C-X-C motif) ligand; IFN: interferon; IL: interleukin; MIS-C: a multisystem inflammatory syndrome in children; TNF: tumor necrosis factor.

## Data Availability

No new data were created.

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
