# Peer review of "Immunology of Multisystem Inflammatory Syndrome after COVID-19 in Children: A Review of the Current Evidence"

_ijms, 2023, doi:10.3390/ijms24065711_

Round 1

Reviewer 1 Report

Filippatos et al in a review titled ` Immunology of Multisystem Inflammatory Syndrome after COVID-19 in Children: a review of current evidence` described in detail what is known so far on innate, humoral and cell immunity in children population with MIS-C developed after COVID-19.

The review is well written and structured with a good quality of the table and figre.

The good point of the review is that authors stress the facts which aspects of the immune response still haven`t been studied.

They also used a good amount of the references to compare the differences between MIS-C and Kawasaki syndrome in children.

The only suggestion would be maybe to compare the cases from different world regions, different virus wave periods, before and after the vaccination was approved for children.

Author Response

Reviewer #1:

Filippatos et al in a review titled ` Immunology of Multisystem Inflammatory Syndrome after COVID-19 in Children: a review of current evidence` described in detail what is known so far on innate, humoral and cell immunity in children population with MIS-C developed after COVID-19.

The review is well written and structured with a good quality of the table and figre.

The good point of the review is that authors stress the facts which aspects of the immune response still haven`t been studied.

They also used a good amount of the references to compare the differences between MIS-C and Kawasaki syndrome in children.

Reply: We would like to thank the reviewer for these good comments.

The only suggestion would be maybe to compare the cases from different world regions, different virus wave periods, before and after the vaccination was approved for children.

Reply: We would like to thank the reviewer for this well-aimed comment. In line with this suggestion, we added an extra paragraph (lines 71-97) regarding MIS-C in different world regions and virus wave periods before and after pediatric vaccination.

Reviewer 2 Report

Dear authors,

Thank you for submitting your manuscript to IJMS. The article fits into the scope of works published in this journal and the title is adequate to the content of the article. This article consists of a total of 20 pages, including 1 table, 1 figure, and a list of a total of 98 literature references.

A few suggestions should be considered:

1.      It might be advisable that the authors consider distributing the entire text as Justify.

2.      It is recommended to place the section ‘Abbreviations’ (page 3) right under the table as a ‘Legend’

3.      Please insert normal laboratory values for the markers cited on page 4, lines 90-92.

4.      Because an exact differential diagnosis is required, it is necessary to mark the differences encountered in MIS-C compared to Kawasaki disease in a dedicated table.

5.      Is figure 1 original?

6.      For better continuity of the research, blank pages should be avoided, therefore page 12 should be revised.

I would like to congratulate the authors for their good work on this review and for their effort to ensure readers a better view of this topic.

Author Response

Reviewer #2:

Dear authors,

Thank you for submitting your manuscript to IJMS. The article fits into the scope of works published in this journal and the title is adequate to the content of the article. This article consists of a total of 20 pages, including 1 table, 1 figure, and a list of a total of 98 literature references.

Reply: We would like to thank the reviewer for this comment.

A few suggestions should be considered:

  1. It might be advisable that the authors consider distributing the entire text as Justify.

Reply: In line with the reviewer’s suggestion, we modified the structure of the text accordingly.

  1. It is recommended to place the section ‘Abbreviations’ (page 3) right under the table as a ‘Legend’

Reply: We modified the section in Tables 1 and 2 according to the reviewer’s suggestion (lines 107 and 129).

  1. Please insert normal laboratory values for the markers cited on page 4, lines 90-92.

Reply: We thank the reviewer for this comment. We added the normal laboratory values of CRP, neutrophil and lymphocyte counts according to the reviewer’s suggestion (lines 116-120).

  1. Because an exact differential diagnosis is required, it is necessary to mark the differences encountered in MIS-C compared to Kawasaki disease in a dedicated table.

Reply: We would like to thank the reviewer for this comment. In line with this suggestion, we created a second table (Table 2) regarding the main differences between MIS-C and Kawasaki disease that are described in the manuscript.

  1. Is figure 1 original?

Reply: We confirm that figure 1 is original. The figure was originally designed by the Biorender in silico tool (https:// biorender.com). We also added this description in Figure 1 legend (lines 361-362).

  1. For better continuity of the research, blank pages should be avoided, therefore page 12 should be revised.

Reply: We totally agree with the reviewer’s suggestion. We modified the text accordingly.

I would like to congratulate the authors for their good work on this review and for their effort to ensure readers a better view of this topic.

Reply: We would like to thank the respected reviewer for all the well-aimed comments and suggestions. We believe that these additions could improve the quality of our manuscript.

Reviewer 3 Report

The authors goal is to give a overview of the expanding understanding of the immune response to SARS-CoV-2 in children.  The paper is divided into introduction & definition and then by sequential elements of the human immune system.  My major critique of this is the overall way it is presented.  Each subsection reads like a list without a coherent summary.  Listing the various changes or maladaptations that has been described should be augmented by a summarizing synopsis. This could be at the beginning or end of each section.  

For example, Page 6 lines 184-189 could be placed at the beginning of the section as summary. The same is true for the description of superantigens in the first full paragraph on page 9; at the beginning of the section would give this more readability.  The final paragraph beginning on that page would best be placed as the last in the section.

Part of the question regarding MIS-C is how it is distinct, in presentation and pathophysiology, from acute infection with SARS-CoV-2.  To this point, children specific date regarding case and/or hospitalization rates would be helpful.

Additionally, page 8 paragraph lines 308-320.  I am unsure why the single study described merits an entire paragraph while this is true for no other study cited in the manuscript.  I do note the author last names are the same although cannot confirm them as the same person.

Author Response

Reviewer #3:

The authors goal is to give a overview of the expanding understanding of the immune response to SARS-CoV-2 in children.  The paper is divided into introduction & definition and then by sequential elements of the human immune system.  My major critique of this is the overall way it is presented.  Each subsection reads like a list without a coherent summary.  Listing the various changes or maladaptations that has been described should be augmented by a summarizing synopsis. This could be at the beginning or end of each section.  

Reply: We would like to thank the reviewer for this important comment. We totally agree, therefore we reconstructed the structure to add a short synopsis at the beginning of innate immune response (lines 100-103), humoral immune response (lines 205-211), cellular immune response section (lines 347-350) and superantigen section (lines 353-357 and 363-367).

For example, Page 6 lines 184-189 could be placed at the beginning of the section as summary.

Reply: We agree with the reviewer. In line with the previous comment, we modified the text according to the reviewer’s suggestion (lines 205-208).

The same is true for the description of superantigens in the first full paragraph on page 9; at the beginning of the section would give this more readability. 

Reply: We totally agree with the reviewer and we modified the text accordingly (lines 353-357 and 363-367).

The final paragraph beginning on that page would best be placed as the last in the section.

Reply: We modified the text according to the reviewer’s comment (lines 403-409).

Part of the question regarding MIS-C is how it is distinct, in presentation and pathophysiology, from acute infection with SARS-CoV-2.  To this point, children specific date regarding case and/or hospitalization rates would be helpful.

Reply: We would like to thank the respected reviewer for this well-aimed suggestion. In line with the Reviewers’ #1 and #3 suggestions, we added an extra paragraph (lines 71-97) regarding the incidence and hospitalization rates in different world regions and SARS-CoV-2 predominance sublineages.

Additionally, page 8 paragraph lines 308-320.  I am unsure why the single study described merits an entire paragraph while this is true for no other study cited in the manuscript.  I do note the author last names are the same although cannot confirm them as the same person.

Reply: We agree with the reviewer’s comment. We agree with the reviewer’s comment, and we condensed the content of the paragraph in one sentence (lines 341-346). This study was conducted by our research group and has been published in Open Forum Infectious Diseases Journal (F. Filippatos et al., “1093. Multiparametric Investigation Of Spike-Protein Specific T-cell Cytokine Expression Profile In Children With Symptomatic COVID-19 Or Multisystem Inflammatory Syndrome,” Open Forum Infect Dis, vol. 9, no. Supplement_2, p. 1093, Dec. 2022, doi: 10.1093/OFID/OFAC492.933).